# Mn-Containing Bioactive Glass-Ceramics: BMP-2-Mimetic Peptide Covalent Grafting Boosts Human-Osteoblast Proliferation and Mineral Deposition

**DOI:** 10.3390/ma15134647

**Published:** 2022-07-01

**Authors:** Leonardo Cassari, Paola Brun, Michele Di Foggia, Paola Taddei, Annj Zamuner, Antonella Pasquato, Adriana De Stefanis, Veronica Valentini, Vicentiu Mircea Saceleanu, Julietta V. Rau, Monica Dettin

**Affiliations:** 1Department of Industrial Engineering, University of Padova, Via Marzolo 9, 35131 Padova, Italy; leonardo.cassari@phd.unipd.it (L.C.); annj.zamuner@unipd.it (A.Z.); antonella.pasquato@unipd.it (A.P.); 2Department of Molecular Medicine, University of Padova, Via Gabelli, 63, 35121 Padova, Italy; paola.brun.1@unipd.it; 3Department of Biomedical and Neuromotor Sciences, University of Bologna, Via Irnerio 48, 40126 Bologna, Italy; michele.difoggia2@unibo.it (M.D.F.); paola.taddei@unibo.it (P.T.); 4Istituto di Struttura della Materia, Consiglio Nazionale delle Ricerche (ISM-CNR), Montelibretti Unit, Via Salaria km 29.300, Monterotondo, 00015 Rome, Italy; adriana.destefanis@ism.cnr.it (A.D.S.); veronica.valentini@ism.cnr.it (V.V.); 5Faculty of Medicine, University Lucian Blaga Sibiu, 2A Lucian Blaga Street, 550169 Sibiu, Romania; vicentiu.saceleanu@gmail.com; 6Istituto di Struttura della Materia, Consiglio Nazionale delle Ricerche (ISM-CNR), Via del Fosso del Cavaliere, 100, 00133 Rome, Italy; giulietta.rau@ism.cnr.it; 7Department of Analytical, Physical and Colloid Chemistry, Sechenov First Moscow State Medical University, Trubetskaya 8, Build. 2, 119991 Moscow, Russia

**Keywords:** BMP-2, bone tissue engineering, glass-ceramic, covalent functionalization

## Abstract

The addition of Mn in bioceramic formulation is gaining interest in the field of bone implants. Mn activates human osteoblast (h-osteoblast) integrins, enhancing cell proliferation with a dose-dependent effect, whereas Mn-enriched glasses induce inhibition of Gram-negative or Gram-positive bacteria and fungi. In an effort to further optimize Mn-containing scaffolds’ beneficial interaction with h-osteoblasts, a selective and specific covalent functionalization with a bioactive peptide was carried out. The anchoring of a peptide, mapped on the BMP-2 wrist epitope, to the scaffold was performed by a reaction between an aldehyde group of the peptide and the aminic groups of silanized Mn-containing bioceramic. SEM-EDX, FT-IR, and Raman studies confirmed the presence of the peptide grafted onto the scaffold. In in vitro assays, a significant improvement in h-osteoblast proliferation, gene expression, and calcium salt deposition after 7 days was detected in the functionalized Mn-containing bioceramic compared to the controls.

## 1. Introduction

Manganese (Mn) is an essential micronutrient element with therapeutic properties for the human body [1,2]. It is present in various mammal tissues [3] and plays multiple roles in the body. It is a cofactor for several enzymes, such as manganese superoxide dismutase, arginase, and pyruvate carboxylase [4]. Through these enzymes, Mn is involved in cholesterol, glucose, and carbohydrate metabolism; bone formation; reproduction; and immune response [5,6,7].

Mn has been shown to activate h-osteoblast integrins, improving cell adhesion, proliferation, and spreading [8], and its effect on cell functions is dose dependent [9,10].

Mn has recently attracted attention for inclusion in bioceramic bulk materials and coatings [11,12]. Mn-containing β-tricalcium phosphate (Mn-TCP) significantly improves the scavenging of oxygen and nitrogen radicals, exhibiting antioxidant properties [13]. The authors’ findings [13] revealed that Mn ions released from bioceramics inhibit the formation of osteoclasts, promote the differentiation of osteoblasts, and improve bone regeneration under in vivo osteoporotic conditions by ROS scavenging.

Bioceramic materials, such as bioactive glasses containing Mn, have received special attention from the research community, and recently, several groups have proposed several bioactive glass compositions containing Mn, employing various preparation processes [14,15,16,17,18,19,20,21,22,23]. All the reported formulations have been characterized by good in vitro activity assessments. Indeed, Mn addition induced growth and differentiation of different cell lines, such as human mesenchymal stem cells (hMSCs) [8,24], osteoblast-like cells MG-63 [14,20], human bone marrow-derived mesenchymal stromal cells (BMSCs) [21], and equine adipose tissue-derived mesenchymal stem cells (AMSCs) [22]. In this latter case, the presence of Mn favored the stem cell differentiation in adipogenic, chondrogenic, and osteogenic lineages, with the last one being more pronounced [22].

Regarding potential antibacterial characteristics, authors [17] reported that Mn-containing mesoporous bioactive glass exhibited a good antibacterial effect against *Bacillus subtilis* (*B. subtilis*), *Pseudomonas aeruginosa* (*P. aeruginosa*), and *Staphylococcus aureus* (*S. aureus*). On the contrary, in [22], the prepared Mn bioactive glass material induced inhibition of Gram-negative *Salmonella typhi* (*S. typhi*) and *Escherichia coli* (*E. coli*), Gram-positive *Enterococcus faecalis* (*E. faecalis*), and the tested fungus *Candida albicans* (*C. albicans*), while the effect on *P. aeruginosa* and *S. aureus* was not significant.

In a recent review dedicated to silicate-based nanoceramics in regenerative medicine [25], various applications of bioactive glasses were described. Besides the above-described bone regeneration properties and antibacterial applications in bone-related materials, several recent publications have proposed applying Mn-containing bioactive glasses for hyperthermia therapy against tumors [26,27]. Another possible employment for Mn-doped nano bioactive glass is related to scaffolds for contrast enhancement in magnetic resonance imaging [24]. In a recent article [10], the antioxidant effect of Mn-doped bioactive glasses has been discussed. This effect can be attributed to the ability of Mn to extinguish peroxyl radicals into their respective non-radical forms. Moreover, Mn ions can replace Fe ions as a cofactor in enzymes at risk of oxidative attack, enhancing oxidative stress resistance.

In bone tissue engineering research, the study of finely tuned bioceramic composition could be coupled to the bioceramic surface modification with bioactive molecules able to stimulate h-osteoblast adhesion and proliferation. Functionalization of surfaces with Mn improves the biological performance of orthopedic implants such as cell adhesion, proliferation, and osseointegration. Indeed, high levels of Mn increased gene expression of osteoblast-related genes such as Runt-related transcription factor 2 (RUNX2) [28]. The expression of RUNX2 gene plays a critical role in osteoblast maturation and differentiation into cells able to produce osteopontin, osteonectin (SPARC), and vitronectin (VTN) [29]. Expression of SPARC was reported during osteogenic differentiation of mesenchymal stem cells. Even if SPARC is a glycoprotein expressed in cell populations undergoing remodeling (fibroblasts, osteoblasts, and chondrocytes), it is considered a marker of osteogenic progression and adhesion as it is involved in binding type I collagen [30,31]. Moreover, RUNX2 and SPARC promote mineral crystal formation and mineralization. Finally, VTN is a glycoprotein of the extracellular matrix that drives collagen fiber formation and reverses bone loss, thus promoting osteogenic differentiation [32,33]. Many different methods for the delivery of bioactive sequences into biomaterials have been reported in the literature (adsorption, release from coating, covalent anchorage) [34]. However, the adhesive sequences should be covalently anchored [35] to the biomaterial in order to avoid their premature release and promote a more stable cellular adhesion structure through cell mechanosensor recognition [36].

In an effort to further increase Mn-containing scaffold potentiality in bone tissue engineering, a selective and specific covalent functionalization [37,38] of Mn-containing bioactive glass (composition: Na_2_O-K_2_O-MgO-MnO-CaO-CaF_2_-P_2_O_5_-SiO_2_, named MnGC) with a bioactive peptide was performed. The anchored bioactive sequence is a BMP-2-mimicking peptide (sequence 48–69 of BMP-2 wrist epitope) that has been proven to guide osteogenic differentiation of Mesenchymal Stem Cells [39] and promote h-osteoblasts’ proliferation; calcium deposition; and gene expression of Runx2, Vitronectin, and Sparc when conjugated to a porous chitosan scaffold [40]. The present study demonstrated that the combined modification of both bioactive glass-ceramic composition and BMP-2-mimicking peptide (here named BMP_aldehyde) enrichment could act synergistically to improve bone biomaterial performance.

## 2. Materials and Methods

### 2.1. Materials

The solid support Rink Amide mBHA resin, all Fluorenylmethyloxycarbonyl (Fmoc) protected amino acids, 2-(1H-benzotriazol-1-yl)-1,1,3,3-tetramethyluronium hexafluoro phosphate (HBTU), and ethyl cyano(hydroxyimino)acetate (Oxyma Pure) were obtained from Merck Millipore (Burlington, MA, USA). *N*,*N*-Diisopropylethylamine (DIPEA), Piperidine, and Trifluoroacetic acid (TFA) were from Biosolve (Valkenswaard, Holland). Solvents for synthesis such as *N*,*N*-dimethylformamide (DMF) and Dichloromethane (DCM) were obtained from Merck Millipore. *N*-methyl-2-pyrrolidone (NMP) was obtained from Iris Biotech GmbH (Marktredwitz, Germany). Triethylsilane (TES), (3-Aminopropyl)triethoxysilane (APTES 2%), and Acetone were obtained from Merck Millipore. Solvents for chromatography such as acetonitrile and TFA (HPLC grade) were obtained from Sigma-Aldrich. Sodium cyanoborohydride (NaCNBH_3_) was obtained from Millipore Corporation (Merck KGaA, Darmstadt, Germany), and Sodium periodate from Sigma-Aldrich.

All products utilized for the glass-ceramic preparation (Nitric acid (HNO_3_), Tetraethylorthosilicate (TEOS), Triethylphosphite (P(OEt)_3_), Calcium nitrate tetrahydrate (Ca(NO_3_)_2_·4H_2_O), Sodium nitrate (NaNO_3_), Magnesium nitrate hexahydrate (Mg(NO_3_)_2_·6H_2_O), Ammonium fluoride (NH_4_F), Manganese(II) nitrate tetrahydrate (Mn(NO_3_)_2_·4H_2_O), and Potassium nitrate (KNO_3_)) were obtained from Sigma Aldrich, with purity ≥ 98%.

### 2.2. Bioceramic

#### 2.2.1. Bioceramic Preparation

MnGC (Na_2_O [2.00]-K_2_O [0.09]-MnO [2.07]-CaO [38.78]-CaF_2_·[2.25]-P_2_O_5_·[9.66]-SiO_2_ [41.71][wt%]) glass-ceramic was prepared by sol-gel method according to the procedure described in [21].

To a 0.1 M nitric acid aqueous solution, all the other components were added one by one, under vigorous stirring, as per the order given in Section 2.1, except for NH_4_F and Mn(NO_3_)_2_·4H_2_O, which were added together. Products were added at intervals of 30 min. Nitric acid was used to catalyze TEOS and hydrolyze P(OEt)_3_. The so-obtained sol was kept for 10 days at room temperature, without stirring, and then stored in an oven at 70 °C for 72 h to transform the sol into the gel. The gel was dried for 48 h at 120 °C and then treated in a furnace at 700 °C (peak T) in air. The resulting granules were sintered in air at 1100 °C (peak T). Finally, the product was finely ground and sieved. For testing purposes, disks (thickness 3 mm, ∅ 10 mm) were obtained by pressing the powder at 500 MPa for 5 min with a disc press.

#### 2.2.2. Bioceramic Characterization

X-ray powder diffraction (XRPD) patterns of the solids were recorded on a Philips diffractometer (model PW 1130/00) using Ni-filtered CuK radiation (λ = 1.5418 Å) at 40 Kv and 20 mA. Data were collected from 20° to 50° 2θ at a rate of 1° 2θ/min with a step width of 0.06° 2θ. Both minerals were identified according to their American Mineralogist Crystal Structure Database.

Thermogravimetry/differential thermal analysis (TG-DTA) was performed on a Stanton TG-DTA 1500 (scan range 30–1150 °C; scan rate 10 °C min^−1^, flow gas: air).

UV-vis spectra (as Diffuse Reflectance Spectroscopy), in the range 200–800 nm, were registered on a Perkin Elmer Lambda 9 instrument equipped with a 60 mm integrating sphere, against MgO as a reference. Data were collected at an interval of 0.2 nm and 60 nm/min rate with a slit of 2 nm in the UV-vis range.

Raman measurements were carried out using a Horiba Scientific LabRam HR Evolution confocal spectrometer equipped with a 17 mW He/Ne laser source (λ_exc_ = 633 nm), an electron-multiplier CCD detector, and an Olympus U5RE2 microscope. A 100× objective with a numerical aperture (NA) of 0.9 and a grating with 600 grooves/mm was used, and the duration of the spectra collection was 5 s per 10 accumulations at 100% of laser power on the sample.

### 2.3. Peptide

#### 2.3.1. Peptide Synthesis

The peptide H–Ser–x–Pro–Phe–Pro–Leu–Ala–Asp–His–Leu–Asn–Ser–Thr–Asn–His–Ala–Ile–Val–Gln–Thr–Leu–Val–Asn–Ser–NH_2_, where *x* represents 7-amino heptanoic acid, was synthesized by standard Fmoc chemistry using Rink-Amide mBHA resin (substitution 0.52 g/mmol); 240 mg of resin (corresponding to 0.125 mmol) was used. The peptide was obtained through a solid-phase synthesis process using the Syro I automatic synthesizer (Multisyntech GmbH, Witten, Germany). The side-chain protections were tBu for Ser and Thr; OtBu for Asp; and Trt for His, Asn, and Gln. Five equivalents of each amino acid, five equivalents of activating agent HBTU/Oxima Pure, and ten equivalents of DIPEA with respect to the resin reactive groups were used for each coupling of 45 min. Peptide cleavage from the resin and the contemporary side chain groups deprotection were carried out by adding, in the following order, 0.125 mL of MilliQ water, 0.125 mL of TES, and 4.725 of TFA mL at room temperature for 90 min.

#### 2.3.2. Peptide Oxidation

The oxidative reaction of N-terminal Ser of the above-reported peptide with sodium periodate (2.5 mM NaIO_4_ in water) produces an N-terminal aldehyde group. The reaction was carried out at room temperature for 4 min.

After the oxidation, the crude BMP_aldehyde was purified through reverse-phase semipreparative chromatography under the following conditions: column, Zorbax 300SB C_18_ (5 µm, 300 Å, 9.4 × 250 mm); Eluent A, 0.05% TFA in MilliQ water; Eluent B: 0.05% TFA in CH_3_CN; gradient, from 0% B to 22% B in 2 min, and from 22% B to 37% B in 60 min; flow rate, 4 mL/min; detector at 214 nm.

#### 2.3.3. Peptide Characterization

The chromatogram of the purified peptide BMP_aldehyde was carried out under the following conditions: column, Vydac 218TP C_18_ (5 µm, 300 Å, 4.6 × 250 mm); Eluent A, 0.05% TFA in MilliQ water; Eluent B: 0.05% TFA in CH_3_CN; gradient, from 22% B to 44% B in 40 min; flow rate, 1 mL/min; detection at 214 nm; injection volume, 30 µL. The retention time was 23.5 min, and the purity grade was 99% (data not shown). The MALDI mass analysis confirmed the product’s identity (Experimental mass: 2571.09 Da, Theoretical mass: 2570.94 Da (4800 MALDI-TOF/TOF TM analyzer provided with 4000 Series Explorer TM software, Applied Biosystem/MDS Sciex, Foster City, CA, USA)).

### 2.4. Functionalized Bioceramic

#### 2.4.1. Functionalization

The MnGC samples were immersed in a 2% APTES/acetone solution. The reaction took place at 40 °C overnight. The samples underwent the following washings for 1 min each: DCM (three times), Acetone (three times), and H_2_O MilliQ (three times). The silanized MnGC was dried at 100 °C for 10 min.

Scaffolds were functionalized with a 1 mg/mL solution of BMP_aldehyde in 10% DMF/H_2_O MilliQ with the addition of NaCNBH_3_ (3 mg/mL) as a reducing agent. The reaction took place overnight at room temperature. The product of the reaction, BMP-MnGC, which is MnGC functionalized with BMP_aldehyde and reduced, underwent three washings with MilliQ water and was then dried under vacuum for 1 h.

#### 2.4.2. Characterization of the Functionalized Scaffold

Raman spectra were registered on a Jasco NRS-2000C spectrometer equipped with a microscope with 100× magnification. The spectra were recorded under backscattering conditions with 4 cm^−1^ spectral resolution using the 532 nm line (DPSS laser driver, LGBlase LLC, Fremont, CA, USA) with a power of about 20 mW. The detector was a liquid nitrogen cooled CCD (Spec-10: 100B, Roper Scientific, Inc., Tucson, AZ, USA). Each spectrum was the average of 16 measurements.

IR spectra were recorded with a Bruker ALPHA series FT-IR spectrometer (Bruker, Ettlingen, Germany) equipped with an attenuated total reflectance (diamond crystal) apparatus and a Deuterated Lanthanum α-Alanine doped TriGlycine Sulfate (DLaTGS) detector. The spectra were averaged over 100 scans at a resolution of 4 cm^−1^.

The morphology of both MnGC and BMP-MnGC has been investigated by a Scanning Electron Microscopy (SEM). The machine employed is a COXEM EM 30AX plus equipped with a Tungsten Filament (W), a SE Detector, and BSE Detector (Solid type 4 Channel).

The success of the functionalization process has been tested through an elemental analysis with SEM-EDX, i.e., the aforementioned SEM equipped with an energy dispersive X-ray detector (EDX, model EDAX Element-C2B).

### 2.5. Biological Assays

#### 2.5.1. Cell Culture

Human (h) osteoblast cells were obtained from explants of cortical mandible bone collected during a surgical procedure from a 45-year-old healthy man. The bone fragments were cultured at 37 °C in 5% CO_2_ and 95% humidity in D-MEM/F12 (1:1) medium supplemented with 20% *v/v* heat-inactivated fetal bovine serum, 1% *v/v* sodium pyruvate, 1% *v/v* nonessential amino acids, 1% *v/v* antibiotic–antimycotic solution, and 1 U/mL insulin (complete medium; all reagents were provided by Gibco, Invitrogen, Milan, Italy). Cultures were periodically checked for contamination and carried on until cells became confluent. Cells were then detached using trypsin EDTA (Gibco) and cultured in a complete medium supplemented with 50 mg/mL ascorbic acid, 10 nM dexamethasone, and 10 mM β-glycerophosphate (all purchased from Merck). After ten days of culture in the supplemented medium, cells were subjected to alkaline phosphatase (ALPL) activity and von Kossa staining; the purity of the culture was greater than 96%. The study was approved by the Ethical Committee of the University Hospital of Padova; the patient was informed about the study’s aims and protocol and gave his written informed consent.

At the time of the experiments, cells were detached using Trypsin-EDTA, washed, and resuspended in a complete supplemented medium at a density of 1 × 10^5^ cells/mL. Cells were placed on the surface of glass matrices (1 cm^2^) and cultured at 37 °C in 5% CO_2_ and 95% humidity. Culture media were renewed every two days; pH changes of the media were periodically inspected.

#### 2.5.2. Proliferation Assay

The proliferation of osteoblasts cultured on functionalized and non-functionalized glass-ceramics (i.e., MnGC and BMP-MnGC) was assessed at 24 h or 7 days using carboxyfluorescein diacetate succinimidyl ester CFSE (Molecular Probe, Invitrogen), a cell-permeable fluorescent probe equally partitioned among daughter cells. Cells were first incubated with CFSE 25 µM at 37 °C for 10 min in pre-warmed PBS containing 0.1% *v/v* BSA. The reaction was stopped by adding 5 volumes of ice-cold culture media. Cells were then washed, counted using Trypan blue, and seeded on glass-ceramics. At the end of incubation, cellular monolayers were washed in PBS and incubated for 5 min with trypsin-EDTA. Cell proliferation was assessed using BD FACS-Calibur flow cytometer by evaluating the percentage of CFSE-positive cells in 10,000 events.

#### 2.5.3. Quantitative Real-Time Polymerase Chain Reaction

Specific mRNA transcript levels coding human Runt-related transcription factor 2 (*RUNX2*), Vitronectin (*VTN*), and Osteonectin (*SPARC*) were quantified in osteoblast cells cultured for 24 h on functionalized and non-functionalized glass-ceramics. At the end of incubation, the total RNA was extracted using SV Total RNA Isolation System kit (Promega, Milan, Italy). Contaminating DNA was removed by DNase I digestion (Omega Bio-Tek, Norcross, GA, USA). cDNA synthesis and subsequent polymerization were performed in one step using the iTaq Universal SYBR Green One-Step Kit (Bio-Rad, Hercules, CA, USA). The reaction mixture contained a 200 nM forward primer, 200 nM reverse primer, iTaq universal SyBR Green reaction mix, iScript reverse transcriptase, and a 200 ng total RNA. Real-time PCR was performed using the ABI PRISM 7700 Sequence Detection System (Applied Biosystems, Milan, Italy). Data were quantified by the ∆∆CT method using human *GAPDH* as the reference gene. Target and reference genes were amplified with efficiencies near 100%. The oligonucleotides used for PCR are listed in Table 1.

#### 2.5.4. Alizarin Staining

Osteoblast cells cultured on functionalized or non-functionalized Mn-containing glass-ceramics were washed two times with PBS and fixed using 10% PFA for 30 min. The cells were then stained with 40 mM Alizarin red (pH 4.2) for 40 min in the dark at room temperature. The cells were incubated at −20 °C for 30 min. Stained samples were examined using a Leica DM4500B microscope (Leica Microsystems, Weitzlar, Germany). In a parallel set of experiments, stained samples were lysed in acetic acid 10% *v*/*v*. The samples were then incubated at 85 °C for 10 min and centrifuged. The pH of the supernatants was neutralized before reading the absorbance of Alizarin red at 405 nm using a microplate reading (Tecan) [8].

#### 2.5.5. Statistical Analysis

Biological data are reported as mean ± standard error. Statistical analysis was performed using the One-way ANOVA test followed by Bonferroni’s multicomparison test, using Graph-Pad Prism 3.03. *p*-values < 0.05 were considered statistically significant.

## 3. Results

### 3.1. Bioceramic Characterization

#### 3.1.1. X-ray Powder Diffraction

XRPD patterns of Mn-doped bioactive glass-ceramic after treatment at 120, 700, and 1100 °C were recorded. It was possible to identify Wollastonite and Fluoroapatite contributions both at 700 (partially) and 1100 °C (Figure 1). Wollastonite is a calcium silicate mineral (CaSiO_3_) and Fluoroapatite is a calcium fluoro phosphate (Ca_5_(PO_4_)_3_F); both minerals were identified according to the American Mineralogist Crystal Structure Database [41,42].

#### 3.1.2. TG-DTA

Mn-doped bioactive glass-ceramic was analyzed as a gel obtained at 120 °C by recording its weight changes and enthalpies (Figure 2). According to the literature on Mn-doped glasses [43], the weight increase can be attributed to transformations of manganese oxides (and possibly hydroxides) due to oxidations in air. In particular, conversion of Mn_3_O_4_ to α-Mn_2_O_3_ was identified, with weight gain, by Pattanayak et al. [44]. It is, in fact, reported that such increases do not occur when samples are thermally treated under N_2_. The subsequent weight loss at higher temperatures can be attributed to the inverse phenomenon of Mn oxides decomposition.

#### 3.1.3. UV-Vis Spectroscopy

Mn-doped bioactive glass-ceramic was analyzed as a gel at 120 °C and after treatment at 700 and 1100 °C (Figure 3). It was possible to identify both Mn(II) and Mn(III), whose electronic transitions were assigned according to the literature [45,46,47] (Table 2).

#### 3.1.4. Raman Spectroscopy

As in the case of the X-ray characterization in the Raman spectra, it was possible to identify Wollastonite and Fluoroapatite contributions both at 700 (partially) and 1100 °C (Figure 4). Frequencies and assignments of bands for both samples are given in Table 3.

### 3.2. Bioceramic Functionalization with BMP_aldehyde

The bioceramic silanization with APTES, reported in Section 2.4.1, modified the surface of bioceramic samples, introducing useful amino groups. The bioactive peptide anchoring was carried out through BMP_aldehyde able to react with surface -NH_2_ groups to give a Schiff’s base, and then reduced (Figure 1). The reduction of imines to amines allows stabilization of the grafting bonds; in fact, the reaction to synthesize imines is reversible, and the reverse reaction causes the release of peptides from the glass-ceramic surface. The condensation was carried out under mild conditions (room temperature, aqueous solution), preserving the peptide structure. In addition, this covalent anchoring strategy assures the reaction of only the aldehyde group of the peptide with the surface, preserving the side chain functional groups necessary for cell bindings.

### 3.3. Raman Spectroscopy

The Raman spectrum of BMP-MnGC (Figure 5) is dominated by the bands of aromatic amino acids such as Phenylalanine (Phe) and Histidine (His) [53]. This result suggests that bands are enhanced by the proximity to the Mn-containing glass-ceramic surface, according to the SERS (Surface Enhanced Raman Scattering) effect [54]. The absence of the typical amide I and amide III spectral bands (which are sensitive to the protein secondary structure) is not surprising, as previously described in SERS spectra of native insulin and amyloid fibrils [55,56] and attributed to the orientation and the distance between the peptide bond and the metal surface caused by the presence of bulky amino acids side chains. Some bands assignable to APTES were detected, while no signals assignable to MnGC were observed either in BMP-MnGC or in the unfunctionalized silanized scaffold.

### 3.4. IR Spectroscopy

IR spectroscopy allowed the characterization of the silanized MnGC support (Figure 6). Vibrations due to Si-O and Ca-O bonds were observed, confirming the predominant composition as calcium-silicate; bands assignable to phosphate groups were also detected. The occurrence of silanization may be revealed by the broadening around 1500 cm^−1^, according to other studies [57,58], without excluding the possibility that APTES may contribute to the MnGC broad bands in the range of 1000–800 cm^−1^. The IR spectrum of the BMP-MnGC shows the bands of aromatic amino acids, especially Phe, at 694 cm^−1^. No bands assignable to the MnGC support were detected, suggesting that the thickness of the peptide layer was greater than 2 microns (i.e., the sampling depth of the diamond ATR technique).

Moreover, it may be observed that the different spectra recorded in different areas of the same sample appeared coincident, suggesting that the peptide layer was homogeneous at IR spectroscopy. As previously described for Raman spectra, the absence of the typical amide bands (in particular, the IR intense amide I and II bands) should be attributed to the particular orientation and distance between the peptide bond and the surface, which are critical physical parameters of the IR signal enhancement due to the Mn-doping of the bioceramic. Moreover, Guo et al. [59] reported distortions to IR amide I and amide II bands of an alfa-helical protein adsorbed on Ni/nitriloacetic-modified gold films depending on film thickness and radiation incident angle with a very similar spectroscopic setup (i.e., surface-enhanced infrared absorption with attenuated total reflection apparatus). The above reported spectral features suggest the effective immobilization of the peptide onto the MnGC support; in addition, the appearance of the weak spectral features at 1180 and 1154 cm^−1^ may be interpreted as a sign of covalent bonding according to Figure 1; according to the literature [60], these bands may be assigned to the C-N stretching mode of the secondary amine formed upon the reduction of the Schiff’s base (see Figure 1).

### 3.5. SEM

BMP-MnGC and MnGC scaffolds were investigated by SEM to verify the presence of the peptide and to characterize surface morphology and porosity. Porosity results described an inhomogeneous surface, but we can explain that considering the preparation technique [22], which involves the use of powder with different grains and microstructure (Figure 7).

The presence of the peptide was investigated through an elemental analysis, SEM-EDX (Figure 8), considering the nitrogen percentage on the surface of BMP-MnGC. 

The quantity of detected nitrogen was higher than the instrument detection limit (~1%). This confirms the presence of the peptide on the BMP-MnGC scaffold surface, especially in the dark-grey colored spots with a gelatinous appearance (Figure 7). The absence of nitrogen peak in EDX spectrum of non-functionalized MnGC, reported in Appendix A, confirms that the nitrogen peak, observed in the BMP-MnGC sample, is not an artifact.

### 3.6. BMP Functionalization of MnGC Supports h-Osteoblast Proliferation

Cell proliferation was evaluated in h-osteoblasts cultured for 24 h or 7 days on cell culture plastic supports (CP), MnGC, and BMP-MnGC. As reported in Figure 9, following 24 h of culture, MnGC per se did not significantly enhance cell proliferation compared with plastic support, whereas functionalization with BMP_aldehyde and following reduction increased the proliferation by 3.5 fold compared with non-functionalized MnGC (*p* < 0.02).

Following 7 days in culture, proliferation increased two-fold in cells cultured on plastic supports, whereas cell proliferation did not significantly increase in MnGC or MnGC-BMP compared to data recorded at 24 h. However, MnGC functionalization supported cell proliferation at 7 days in culture. Cells recovered from MnGC-BMP reported a greater CFSE-related fluorescent signal compared with cells cultured on MnGC (% of fluorescence: 19.68 ± 0.52 vs. 7.87 ± 0.78, respectively; *p* < 0.02).

### 3.7. Glass-Ceramics Functionalized with BMP Drive h-Osteoblast Differentiation

Behind cell proliferation, functionalized biomaterials should successfully support cell differentiation to ensure osseointegration and bone healing. The ability of BMP functionalized glass-ceramics to support cell differentiation was evaluated by measuring the mRNA transcript levels of three crucial genes involved in osteoblast differentiation. As reported in Figure 10, following 24 h in culture, both MnGC and MnGC-BMP increased expression of *RUNX2*, *VTN*, and *SPARC* mRNA compared with cells cultured on CP. However, the functionalization of MnGC with BMP_aldehyde and following reduction significantly increased the gene expression compared with MnGC. Indeed, BMP_aldehyde functionalization and following reduction increased by 3.75 fold the expression of RUNX2 coding signal factors for cell proliferation [61], by 1.7 fold the expression of VTN coding extracellular matrix proteins essential in osteogenic differentiation [62,63], and by 3.2 fold the mRNA expression of SPARC, a non-collagenous calcium-binding protein involved in bone development [64].

MnGC-BMP also promoted calcium deposition as observed by alizarin red staining in cells cultured for 7 days. As reported in Figure 11, calcium salts deposition was greater in cells cultured on MnGC-BMP compared with cells cultured on MnGC or CP (*p* < 0.02). We did not observe any statistical differences in calcium deposition between cells cultured on MnGC and CP. Therefore, our data demonstrate the ability of BMP functionalization of MnGC in supporting and maintaining proliferation and differentiation in human primary osteoblast cells.

## 4. Discussion

The prefix “bio” in words such as bioceramics refers to the capacity of the biomaterials to interact positively with cells. In particular, the success of bioactive glasses seems to be intrinsic to their degradation process, where (i) growth factors remain entrapped within the gel phase formed during degradation, (ii) ECM structural proteins form strong bonds with degradation particles, and (iii) osteoblast differentiation is promoted from silicon ions. The novel formulation of bioactive glass reported here presents Mn inclusion; this element has been shown to improve osteoblast adhesion, proliferation, and spreading [7] with a dose-dependent effect [8,9]. In addition, Mn ions released from bioceramics inhibit the formation of osteoclasts, promote the differentiation of osteoblasts, and improve bone regeneration under in vivo osteoporotic conditions by ROS scavenging.

In order to improve cell interaction with bioceramic surfaces, we have realized a surface modification of the MnGC with a BMP-2 mimetic peptide. This peptide, reproducing the sequence 48–69 of the BMP-2 protein, covalently linked to a chitosan porous scaffold [40], has shown improved h-osteoblast adhesion, proliferation, and calcium deposition. Usually, materials modified with the inclusion or grafting of bioactive molecules are defined as bioactive materials because they interact with cells using the biochemical cell language; consequently, we are formulating a bioactive bioceramic. In an effort to consider one modification effect at a time: the MnGC increases h-osteoblast adhesion, proliferation, and calcium salt deposition with respect to the control (tissue culture well surface, optimized for cell adhesion), but the differences are not statistically significant. On the other hand, considering gene expression assay results, the MnGC significantly increases all three considered genes. The presence of anchored BMP-2 mimetic sequence, confirmed by Raman, IR, and SEM-EDX analyses, further improves both adhesion and proliferation of h-osteoblasts. Statistically significant differences are registered in mineralization and gene expression assays in favor of BMP-MnGC compared to non-functionalized MnGC.

Our results demonstrate a positive cell response to glass-ceramic functionalization with this peptide reproducing the BMP-2 wrist epitope as in recent studies where bioceramics were functionalized with whole BMP-2 [65] or linear [66] peptides mapped on knuckle epitope of BMP-2. In the article by Damia et al. [65], ceramic spheres were functionalized with BMP-2 by silanization followed by condensation with NHS-PEG6-maleimide, and then BMP-2. The authors demonstrated that the covalently anchored protein maintains its bioactivity even if the method fails to ensure a specific, selective covalent binding on a single functional group of the protein, and is therefore able to orient the protein in an orderly manner on the surface, as was the case with our strategy. In addition, in the study by Zhou et al. [66], the anchoring of the fragment reproducing the sequence 73–92 of the Knuckle epitope of BMP-2 is non-specific, in contrast to what was proposed in the recent article of Oliver-Cervello et al. [67]. In the latter study, the contemporary anchoring to glass in a 1:1 ratio of two peptides (RGD and the sequence 30–34 of BMP-2), which represent an extracellular matrix biomimetic interface, has led to an increased cell adhesion testifying the possibility of using more appropriately spaced signal peptides to promote specific cellular behavior. In Zamuner et al. [68], wollastonite and diopside bioceramic scaffolds were specifically and selectively functionalized with retro-inverso adhesive peptides, i.e., resistant to enzymatic degradation, with excellent results in both in vitro and in vivo assays; in this case, the peptides were condensed to carboxylic groups, appropriately introduced on the surface of the bioceramic, exclusively through their N-terminal amino group, with all reactive side chains protected. In the present study, the necessity of an organic solvent (DMF) and a strong organic acid (TFA) used to solubilize and deprotect the side-chain protected peptide, respectively [68], was avoided, and a specific and selective functionalization was obtained with a simpler and more convenient method that involves fewer chemical reactions under milder conditions (aqueous solution).

In every case, the specific covalent functionalization is to be preferred as it has the advantage of not modifying the side chains of the residues of the protein/peptide used for the surface modification, which are essential for binding to the cellular receptors [69].

## 5. Conclusions

More and more sophisticated strategies could be designed and carried out to improve and direct specific cell behavior in order to integrate the biomaterials with the host tissues; the present study demonstrates the advantage of coupling two different approaches that are, on one hand, the enrichment of glass-ceramics scaffolds with specific elements and, on the other hand, the covalent grafting of bioactive peptides. These approaches are considered synergic to create new generation bone tissue scaffolds. In the near future, further studies will be carried out to evaluate the effective grafted peptide quantity and to individuate the anchored peptide concentration able to optimize h-osteoblast beneficial behavior.

## Data Availability

The data presented in this study are available within this article.

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
