# Peer review of "Mn-Containing Bioactive Glass-Ceramics: BMP-2-Mimetic Peptide Covalent Grafting Boosts Human-Osteoblast Proliferation and Mineral Deposition"

_materials, 2022, doi:10.3390/ma15134647_

Round 1

Reviewer 1 Report

This study investigates the effectiveness of BMP-modified Mn containing bioactive ceramics on boosting HOB proliferation and mineral deposition toward bone tissue repair. The entire manuscript was well written with a reasonable organization. The results and discussion can also support the hypotheses and objectives given in the study. A few small issues are present in the manuscript that may require the authors to further address.

1. BMP was covalently anchored to Mn-bioceramics. However, the results from IR and SEM-EDX did not offer adequate information on the formation of covalent bonding. As shown in Scheme 1, are C-N or other bond specific in modification? without using NMR, how do the authors verify the specific covalent bond formation?

2. Mn is easy to be degraded during the biological environment. Did the author test the weight change of bioceramics when the cell culture assays were performed, specifically on day 7? Has the bioceramic samples still held their original shape on day 7? The structure variation of scaffolds may also affect the cell proliferation and mineral deposition.

3. There are some typos and grammatical mistakes present, such as Line 396, a space between what and proposed. Thoroughly read the manuscript again.   

Author Response

Point by point reply to reviewers’ comments

Reviewer 1

This study investigates the effectiveness of BMP-modified Mn containing bioactive ceramics on boosting HOB proliferation and mineral deposition toward bone tissue repair. The entire manuscript was well written with a reasonable organization. The results and discussion can also support the hypotheses and objectives given in the study. A few small issues are present in the manuscript that may require the authors to further address.

  1. BMP was covalently anchored to Mn-bioceramics. However, the results from IR and SEM-EDX did not offer adequate information on the formation of covalent bonding. As shown in Scheme 1, are C-N or other bond specific in modification? without using NMR, how do the authors verify the specific covalent bond formation?

Raman spectrum could not provide further information about covalent bonding formation with the bioceramic surface. This effect was explained with the Mn doping of the calcium-silicate bioglass, which gave the substrate the ability to enhance the Raman and IR intensity of selected bands with a SERS (Surface Enhanced Raman Scattering)/SEIRA (Surface Enhanced Infrared Absorption) like mechanism [reference 54]. This enhancing mechanism is sensitive to the orientation and the proximity of the different functional groups to the surface; therefore, it is not surprising that both the orientation and the distance from the bioceramic surface would not allow both vibrational techniques to give information about the formation of the bond between APTES and the peptide. The IR spectrum could support the presence of the C-N vibration of secondary amines at 1180 and 1154 cm-1. Figure 6 was updated accordingly. Moreover a paragraph supporting this attribution was added at lines 377-382: “The above reported spectral features suggest the effective immobilization of the peptide onto the MnGC support; in addition, the appearance of the weak spectral features at 1180 and 1154 cm-1 may be interpreted as a sign of covalent bonding according to Scheme 1; according to the literature [60], these bands may be assigned to the C-N stretching mode of the secondary amine formed upon the reduction of the Schiff’s base (see Scheme 1)”. In addition, we have collected some opinions among experts of NMR on solid samples: the NMR instrumentation usually disposable in Universities requires the fragmentation of samples for NMR measurements. In our case, the mechanical treatments of glass-ceramics surely would produce temperature increase and consequently sample modifications with the result to detect the chemical composition of a different material with respect to the target one.

  1. Mn is easy to be degraded during the biological environment. Did the author test the weight change of bioceramics when the cell culture assays were performed, specifically on day 7? Has the bioceramic samples still held their original shape on day 7? The structure variation of scaffolds may also affect the cell proliferation and mineral deposition.

As stated by the Reviewer, cell behavior changes with the scaffolds. For this reason, during the biological experiments, we carefully check changes in pH, the volume of the cultures, and the appearance of the scaffolds. We did not observe any differences between 24 hours and 7 days of culture. We did not record the weight of the scaffolds as they were soaked in the culture media and it would have been difficult to have reproducible data.   

  1. There are some typos and grammatical mistakes present, such as Line 396, a space between what and proposed. Thoroughly read the manuscript again. 

We thank the reviewer for the individuation of typos and grammatical mistakes.  The mistakes have been corrected.

Reviewer 2 Report

The manuscript entitled "Mn-containing bioactive glass-ceramics: BMP-2-mimetic peptide covalent grafting boosts human-osteoblast proliferation and mineral deposition” The functionalization of Mn containing ceramic glass was performed with BMP-2 peptide. The functionalized bioactive glass demonstrated interaction with h-osteoblasts and inhibition of gram-negative or gram- 22 positive bacteria, and fungi. 

The following are my suggestions: 1. The laguage of the manuscript should be improved. There are spelling mistakes like line 27 ‘ FT-IT,  2. Line175, sentence MnGC……..in [22] should be moved to 2.2 bioceramic preparation section 3. The comment on the release of the Mn from the ceramic is desired. 4. Line 257, aminic groups can be amino groups 5. Why reduction of C=N was carried out explanation is desired?  6. FT-IR spectra amide carbonyl peak should be visualized. 7. SEM Edax mapping images should be provided  8.  Photos for samples from cell studies should be provided. 9. Conclusion section should be included.  

Author Response

Reviewer 2

The manuscript entitled "Mn-containing bioactive glass-ceramics: BMP-2-mimetic peptide covalent grafting boosts human-osteoblast proliferation and mineral deposition” The functionalization of Mn containing ceramic glass was performed with BMP-2 peptide. The functionalized bioactive glass demonstrated interaction with h-osteoblasts and inhibition of gram-negative or gram- 22 positive bacteria, and fungi. 

The following are my suggestions: 1. The language of the manuscript should be improved. There are spelling mistakes like line 27 ‘ FT-IT,  2. Line175, sentence MnGC……..in [22] should be moved to 2.2 bioceramic preparation section 3. The comment on the release of the Mn from the ceramic is desired. 4. Line 257, aminic groups can be amino groups 5. Why reduction of C=N was carried out explanation is desired?  6. FT-IR spectra amide carbonyl peak should be visualized. 7. SEM Edax mapping images should be provided  8.  Photos for samples from cell studies should be provided. 9. Conclusion section should be included.  

From 1 to 4. All spelling mistakes, highlighted by the reviewer, were corrected.

  1. An explanation of the importance of C=N reduction is reported in paragraph 3.2: “The reduction of imines to amines consents to stabilize the grafting bonds in fact the reaction to synthesize imines is reversible and the reverse reaction causes the release of peptides from the glass-ceramic surface.”
  2. As evidenced in the text, the Mn doping of the calcium-silicate bioglass gave the substrate the ability to enhance the Raman and IR intensity of selected bands with a SERS (Surface Enhanced Raman Scattering)/SEIRA (Surface Enhanced Infrared Absorption) like mechanism [reference 54]. This enhancing mechanism is sensitive to the orientation and the proximity of the different functional groups to the surface. Several authors have described the absence of the typical amide vibrations from SERS or SEIRA spectra of proteins and peptides. Kuroski et al. [doi: 10.1039/c2an36478f] attributed the absence of Amide I band in SERS spectra of native insulin and amyloid fibrils to the bulkiness of amino acids side chains, as also confirmed by Tabatabaei et al. [doi: 10.1039/C7AN00744B], which also hypothesized an influence of the orientation of the protein compared to the surface. Guo et al. [doi: 10.1016/j.chemphys.2012.11.011] reported distortions to infrared Amide I and amide II bands of an alfa-helical protein adsorbed on Ni/nitriloacetic-modified gold films depending on film thickness and radiation incident angle.

Accordingly, two additional paragraphs were inserted into the text at lines 347-351:

The absence of the typical Amide I and amide III spectral bands, sensitive to the protein secondary structure is not surprising, as previously described in SERS spectra of native insulin and amyloid fibrils [55, 56 ] and attributed to the orientation and the distance between the peptide bond and the metal surface caused by the presence of bulky amino acids side chains.

and at lines 369-377:

As previously described for Raman spectra, the absence of the typical amide bands (in particular, the IR intense amide I and II bands) should be attributed to the particular orientation and distance between the peptide bond and the surface, which are critical physical parameters of the IR signal enhancement due to the Mn-doping of the bioceramic. Moreover, Guo et al. [59] reported distortions to infrared Amide I and amide II bands of an alfa-helical protein adsorbed on Ni/nitriloacetic-modified gold films depending on film thickness and radiation incident angle with a very similar spectroscopic setup (i.e., surface-enhanced infrared absorption with attenuated total reflection apparatus).

  1. SEM Edax spectrum of not functionalized sample was added as supplementary material for comparison.
  2. Representative images of cell proliferation (histogram plots) and alizarin (microscope) were included in the revised version of the Manuscript (Figures 9 and 11).
  3. A conclusion section was included, as required by the reviewer.

Reviewer 3 Report

Comments for “Mn-containing bioactive glass-ceramics: BMP-2-mimetic peptide covalent grafting boosts human-osteoblast proliferation and mineral deposition”

In general, authors provide an interesting work may improve the capacity of bone implants. The manuscript is well presented. There are some minor concerns:   

The key concern is about the performance comparison among CP, MnGC and MnGC-BMP (in Figure 5-7). Based on presented results, the MnGC only cannot enhance CFSE positive cells and other measurement significantly, so authors should show the BMP only in their results to prove the compound MnGC-BMP contribute to human-osteoblast proliferation and mineral deposition rather than the ability of BMP.

The functions of Runx2, Vitronectin, and Sparc should be explained more in the introduction to help readers understand why expression level of these three genes were considered as measurements.

Author Response

Reviewer 3

Comments for “Mn-containing bioactive glass-ceramics: BMP-2-mimetic peptide covalent grafting boosts human-osteoblast proliferation and mineral deposition”

 In general, authors provide an interesting work may improve the capacity of bone implants. The manuscript is well presented. There are some minor concerns:   

The key concern is about the performance comparison among CP, MnGC and MnGC-BMP (in Figure 5-7). Based on presented results, the MnGC only cannot enhance CFSE positive cells and other measurement significantly, so authors should show the BMP only in their results to prove the compound MnGC-BMP contribute to human-osteoblast proliferation and mineral deposition rather than the ability of BMP.

 The difference between MnGc and MnGC-BMP is the anchored peptide so it is consequent to attribute significant increases in proliferation, mineral deposition, and gene expression to this modification. In every case, our goal is to study the biological activity of peptides anchored to implant surfaces, not solution factors. Previous works showed the ability of this BMP fragment to increase h-osteoblast proliferation and mineralization when covalently grafted to other materials [Nanomaterials (Basel). 2021 Oct 21;11(11):2784. doi: 10.3390/nano11112784]

The functions of Runx2, Vitronectin, and Sparc should be explained more in the introduction to help readers understand why expression level of these three genes were considered as measurements.

We thank the Reviewer for his/her suggestions. The role of Runx2, Vitronectin, and Sparc was explained in the introduction of the Revised Manuscript.

Reviewer 4 Report

The articles suggested that under in vitro conditions, the combined use of bioactive glass ceramics and BMP-2 peptide significantly improves bone regeneration. The manuscript is well written. However there are few type errors. The following are the revisions suggested to improve the readability of the manuscript:

In abstract, line 27, “SEM-EDX, FT-IT, and Raman studies confirmed the presence 27 of the peptide grafted onto the scaffolds.” Please check the spelling of FTIR.

In section 2.3.1, line 146, unit of TFA added for side chain deprotection is missing.

In section 2.4.1, line 188, check the following sentence “The morphology of both MgGC and BMP-MnGC have been investigated by a Scanning Electron Microscopy (SEM).” The authors misspelled MnGC as MgGC.

In Figure 4, It is advisable to include EDX spectra of non-functionalized MnGC for comparison and to confirm that the nitrogen peak is from BMP-MnGC not artifacts.

In section 3.4, line 307, the dark grey spots from BMP is shown in Figure 3 not Figure 6. Please correct it.

In the present study, the peptide was covalently linked to the substrate. Hence, it is important to check the amount of peptide conjugated to MnGC. It is advised that the authors should check the amount of BMP conjugated on MnGC and their release kinetics in culture medium in order to assure the efficiency of the protocol followed?

The authors should discuss the limitations of the present study and the future outlooks in the discussion part. Also, a separate heading on conclusions should be included in the manuscript.

Author Response

Reviewer 4

The articles suggested that under in vitro conditions, the combined use of bioactive glass ceramics and BMP-2 peptide significantly improves bone regeneration. The manuscript is well written. However there are few type errors. The following are the revisions suggested to improve the readability of the manuscript:

In abstract, line 27, “SEM-EDX, FT-IT, and Raman studies confirmed the presence 27 of the peptide grafted onto the scaffolds.” Please check the spelling of FTIR.

Thank you for your comment. The mistake was removed.

In section 2.3.1, line 146, unit of TFA added for side chain deprotection is missing.

The unit of TFA was added.

In section 2.4.1, line 188, check the following sentence “The morphology of both MgGC and BMP-MnGC have been investigated by a Scanning Electron Microscopy (SEM).” The authors misspelled MnGC as MgGC.

The typo was corrected.

In Figure 4, It is advisable to include EDX spectra of non-functionalized MnGC for comparison and to confirm that the nitrogen peak is from BMP-MnGC not artifacts.

We have introduced an EDX spectrum of non-functionalized MnGC for comparison in supplementary information section.

In section 3.4, line 307, the dark grey spots from BMP is shown in Figure 3 not Figure 6. Please correct it.

The Figure number was corrected.

In the present study, the peptide was covalently linked to the substrate. Hence, it is important to check the amount of peptide conjugated to MnGC. It is advised that the authors should check the amount of BMP conjugated on MnGC and their release kinetics in culture medium in order to assure the efficiency of the protocol followed?

The authors should discuss the limitations of the present study and the future outlooks in the discussion part. Also, a separate heading on conclusions should be included in the manuscript.

The quantitative determination of grafted peptide is difficult to evaluate because the peptide quantity is very low. In future studies, a fluorescent peptide or a radioactive-labeled analog will be used in functionalization allowing the subsequent determination of the grafted peptide quantity. In my opinion, the determination of the released peptide in solution could be a way to prove the total or partial non-covalent nature of the bond between the peptide and the glass ceramics but in the case of a negative result, it could be read as the confirmation of covalent bond formation but it could be also due of strong secondary interactions between the peptide and the surface. We are studying anchoring chemistry with the release of a leaving group that could be detected to prove the grafting yield (for example detection of alcohol released from the reaction of an active ester). 

Future outlooks were introduced in the conclusion section.

Round 2

Reviewer 1 Report

N/A